# Identifying regions for enhanced control of *gambiense* sleeping sickness in the Democratic Republic of Congo

Ching-I Huang [1,2,6 ✉], Ronald E. Crump [1,2,3,6], Paul E. Brown[1,2], Simon E. F. Spencer [1,4], Erick Mwamba Miaka[5], Chansy Shampa[5], Matt J. Keeling [1,2,3] & Kat S. Rock [1,2]

*Gambiense* human African trypanosomiasis (sleeping sickness, gHAT) is a disease targeted for elimination of transmission by 2030. While annual new cases are at a historical minimum, the likelihood of achieving the target is unknown. We utilised modelling to study the impacts of four strategies using currently available interventions, including active and passive screening and vector control, on disease burden and transmission across 168 endemic health zones in the Democratic Republic of the Congo. Median projected years of elimination of transmission show only 98 health zones are on track despite significant reduction in disease burden under medical-only strategies (64 health zones if > 90% certainty required). Blanket coverage with vector control is impractical, but is predicted to reach the target in all heath zones. Utilising projected disease burden under the uniform medical-only strategy, we provide a priority list of health zones for consideration for supplementary vector control alongside medical interventions.

[1] Zeeman Institute for System Biology and Infectious Disease Epidemiology Research, The University of Warwick, Coventry, UK. [2] Mathematics Institute, The University of Warwick, Coventry, UK. [3] The School of Life Sciences, The University of Warwick, Coventry, UK. [4] The Department of Statistics, The University of Warwick, Coventry, UK. [5] Programme National de Lutte contre la Trypanosomiase Humaine Africaine (PNLTHA), Kinshasa, Democratic Republic of the Congo. [6] These authors contributed equally: Ching-I Huang, Ronald E. Crump. ✉email: ching-i.huang@warwick.ac.uk

The pinnacle of success for an infectious disease programme is to drive the disease to eradication, resulting in complete removal of morbidity and mortality, no longer requiring interventions[1]. Of the human diseases targeted for eradication, only one—smallpox—has currently achieved this objective, yet there are several for which this remains the potentially elusive goal (such as polio, Guinea worm, and yaws)[2–5]. Clear lessons that can be learnt from many eradication programmes are: (i) the often slow progress from low to very low case burden, (ii) the ever-increasing effort required per case to tackle remaining infection, and (iii) the question of whether eradication is even epidemiologically or operationally feasible[6].

One step down from eradication is elimination of transmission (EOT) to humans, acknowledging that transmission pockets could persist in non-human animal cycles[7–9]. This goal, arguably, may be almost as challenging and fraught with the same hurdles to overcome as eradication itself. *Gambiense* human African trypanosomiasis (gHAT, sleeping sickness) is one such disease with the EOT goal and within the last decade, it was still known to be extant in 15 countries in the West and Central Africa[10]. This parasitic infection is transmitted to humans via bites by tsetse, with gHAT symptoms typically increasing in severity over several years and leading to death without treatment. Remarkable progress has been made to bring down the case burden across the continent; cases fell to below 10,000 in 2009 for the first time since the most recent epidemic started in the 1970s, and to only 953 in 2018[10–12]. This reduction has sparked optimism that EOT may be possible and the World Health Organization (WHO) has set the goal of EOT by 2030[13,14]. Indeed gHAT fulfils some of the criteria associated with an "eliminable" disease[15]: we have a range of field-proven tools and associated delivery mechanisms as well as means of diagnosis and surveillance[16]. Unlike smallpox, gHAT is not vaccine-preventable, but widespread testing, diagnosis and treatment have worked well to curtail transmission[12,17]. Advances in treatment have transformed the once toxic intravenous treatment regime into an oral cure for most patients, however, confirmation of the parasite is currently still a requirement before the drugs can be administered[18,19]. This means that infected individuals must be identified either by self-presenting at facilities with gHAT diagnostics due to symptoms or by specially-trained mobile screening teams targeting villages with recent case reporting[16]. A complementary intervention of tsetse control has not been used widely to date but has been shown to be effective at rapidly reducing vector populations in some gHAT-endemic regions where it has been implemented[20–22]. The key questions are (1) how much current tools for gHAT can reduce gHAT burden, and (2) whether they are sufficient to reach EOT in the next 10 years, and if so, how expansive might their use have to be to get there.

The Democratic Republic of the Congo (DRC) is the country with the greatest number of reported gHAT cases. Due to the concerted efforts of the national sleeping sickness control programme in the DRC (PNLTHA-DRC), the number of reported cases dropped below 1000 in the country in 2018. However, the DRC still accounted for ~70% of global cases (660 out of 953 cases) in that year[10,12]. Therefore, the DRC is a critical country on which the achievement of EOT by 2030 hinges.

In order to project the trend of gHAT burden and ultimately assess EOT feasibility, this study focuses on quantitative forecasting of gHAT across the endemic health zones in the DRC to examine if, how, and when EOT could be expected under strategies based on currently available tools. Health zones are the administrative units at which public health care is managed and each has a population of between 29,010 and 613,072 people (the median size is 157,338)[23]. Previous DRC-specific predictive modelling studies have provided insights into expected timelines

to EOT in Equateur province[24], and parts of Bandundu province[25–29] under continuation of medical-based strategies with or without vector control. From these studies, it is clear that a one-size-fits-all approach is unlikely to be sufficient to meet this highly ambitious target in the next decade. Although coverage of active screening has been driven by local numbers of cases, additional data-driven guidance could help to further tailor strategy selection.

In this article, we enlarge the geographical scope of previous predictions to include 168 health zones across the whole country, utilising our recently published fitting results[30] to examine the strategies of active screening (AS) with or without supplementary, large-scale vector control (VC) on top of the local passive screening (PS) system to stop gHAT transmission by 2030 in the DRC. These health zones are ones considered to be endemic during the 2000–2016 period, having reported cases in a minimum of 5 years. We aim to identify regions which are likely to be successful in substantially reducing disease burden and achieving local EOT on their current trajectory and ones where enhanced control may be required to meet this target. Furthermore, we provide a priority list of health zones where intensification of strategies is most urgent based on where is expected to experience the greatest disease burden whilst also being unlikely to reach EOT by 2030. A graphical user interface (GUI) to complement this article was set up to provide full-model outputs.

## Results

**Projection trends in different risk settings.** Our gHAT model, a deterministic, mechanistic, SEIRS-type model, was independently fitted to longitudinal human case data from 2000 to 2016 in 168 health zones in the DRC by Markov chain Monte Carlo (MCMC) methods[30]. We used parameter estimates from our previously fitted gHAT model (posteriors are available at https://hatmepp.warwick.ac.uk/fitting/v2/ and simulated active and passive cases in 2000–2016 can be also viewed at https://hatmepp.warwick.ac.uk/projections/v2/) to simulate forward projections in 168 health zones under four strategies: two medical-only strategies which comprise of active and passive screening (MeanAS and 40%AS), and two medical strategies with supplementary vector control from 2020 (MeanAS+VC and 40% AS+VC). The two selected active screening coverage levels are the mean of the last 5 years of data (2012–2016) and 40% of the health zone's population in 2014. Projections were run for 2017–2050 independently with parameter uncertainty in each health zone. Table 1 in "Methods" gives more detailed information on the strategies presented in the main text while results of HistMaxAS and HistMaxAS+VC strategies (historical maximum coverage achieved in AS between 2000 and 2016) and VC sensitivity analyses can be found in Supplementary Note 2: Results and our GUI at https://hatmepp.warwick.ac.uk/projections/v2/. Those health zones with little or no interventions and/or case reporting were excluded from the original model fitting and hence from these projections[30].

Figure 1 shows assumed numbers of people screened and model outputs (i.e. active and passive cases, new infections, and probability of EOT) for the four strategies in two example health zones: Kwamouth (in the former Bandundu province) and Tandala (in the former Equateur province). Both health zones had significant numbers of cases in the early 2000s and still have on-going transmission despite annual AS. Kwamouth, with 1068 reported cases in 2012–2016 (estimated 2015 population of 127,205), falls within WHO's definition of a "high-risk" category for gHAT (1–10 cases/1000 per year averaged over 5 years), while Tandala is only "low-risk" (38 reported cases in 2012–2016 and estimated 2015 population of 274,945 – i.e. 1–10 cases/100,000

**Table 1 Strategies considered for projections (2017–2050).**

| Strategy name | AS coverage from 2017 | VC effectiveness from 2020 | PS coverage from 2017 |
|---|---|---|---|
| MeanAS | Mean (2012–2016) | 0% | Same as 2016 |
| 40%AS | 40% of the population | 0% | Same as 2016 |
| MeanAS+VC | Mean (2012–2016) | 90% for Yasa Bonga; 80% everywhere else | Same as 2016 |
| 40%AS+VC | 40% of the population | 90% for Yasa Bonga; 80% everywhere else | Same as 2016 |

AS active screening, VC vector control, PS passive screening.
VC effectiveness is denoted here by the proportional reduction in tsetse population after 1 year of implementation. Strategies without VC are not considered in Yasa Bonga because VC has been implemented since the middle of 2015.

per year). Historical AS data shows that Kwamouth had substantially higher proportions of people screened than Tandala. Despite very high coverage of AS in Kwamouth, achieving EOT by 2030 is predicted to only be possible when VC is added—the model suggests that transmission will be interrupted completely within 4 years once VC begins. Unlike Kwamouth, Tandala appears extremely likely to achieve EOT by 2030 with the 40%AS strategy and EOT even occurs in 59% of projections under the less intensive MeanAS strategy. Projections under each strategy for each health zone can be found in our GUI at https://hatmepp.warwick.ac.uk/projections/v2/.

**Timelines to, and certainty of, EOT.** The year of elimination of transmission (YEOT) is defined in our modelling framework as the first year that the EOT criterion is met (i.e. the number of new infections is less than one). Health zone maps of the median YEOT under the four strategies are shown in Fig. 2. Using the median value of YEOT, health zones can be classified into three categories: on track (YEOT ≤ 2030), slightly behind schedule (2030 < YEOT ≤ 2040), and greatly behind schedule (YEOT > 2040) to meet the EOT goal. We predict 75 health zones are on track, 28 are slightly behind schedule, and 65 are greatly behind schedule under the MeanAS strategy. Data show low coverage of AS (median AS coverage is 7.7% and lower than 25% AS coverage in 95% health zones) may be responsible for predicted delays in EOT in health zones outside the former Bandundu province. The 40%AS strategy improves YEOT by an average of 2 years (95% CI: [0, 17.3]) and therefore its predictions are less pessimistic: 98 health zones are on track, 24 are slightly behind schedule, and 46 are greatly behind schedule. There are only three health zones, Bolobo, Kwamouth, and Masi Manimba, all in the former Bandundu province that screened more than 40% of their populations on average during 2012–2016. With VC starting in 2020, all health zones are predicted to achieve EOT by 2024. The 40% AS+VC strategy could further bring forward YEOT by up to 2 years (although the 5-year data bins in Fig. 2 obscure this nuance). It is possible for different strategies to have very similar YEOT distributions within a health zone especially if EOT is expected to have already occurred.

The median YEOT provides a point estimate of when to expect EOT but not the degree of certainty that the goal will be met by 2030. The probability of elimination of transmission (PEOT) by 2030, which reflects the distribution of YEOT, captures the uncertainty of model predictions. Consequently, low values of median YEOT cannot guarantee EOT by 2030. One example is Inongo in the former Bandundu province, which has a median YEOT of 2019 but the PEOT by 2030 is <1 under both medical-only strategies. Figure 3 shows PEOT by 2030 in each health zone under four strategies. Three uncertainty categories of model predictions are particularly interesting: EOT is very likely to be met by 2030 if PEOT > 0.9, EOT by 2030 is highly uncertain when

0.3 < PEOT < 0.7, and EOT is very unlikely to be met if PEOT < 0.1. The model predicts that 42 health zones are very likely to meet the goal and 60 are almost certain to miss it under the MeanAS strategy. High uncertainty in EOT is reported in 33 health zones. Despite the distribution of YEOT being shifted forward in general under the 40%AS strategy (22 extra health zones become very likely to meet the goal by 2030), there are still 40 health zones that remain highly uncertain because of their wide YEOT distributions. With VC starting in 2020, a tight distribution of YEOT means EOT by 2030 is extremely likely everywhere even if its median is quite close to 2030.

Case reporting has been the primary but indirect measure for burden and underlying transmission. So one may expect different health zones with the same number of reported cases very likely to have different disease burden, predicted years of EOT and certainty in EOT by 2030. Lusanga and Mosango health zones in the former Bandundu province are geographically connected and both had 13 total reported cases in 2016. Our model predicts EOT to happen in 2029 and 2027 under the 40%AS strategy and the probability of EOT as 60% and 83%, respectively. These differences come from underlying epidemiological variation such as relative risk of high-risk people, tsetse density or time to detection through passive screening (linked to health facility coverage and attendance). The explanations for some of these differences are explored in our fitting paper[30] and the posterior distributions of the parameters can be found in the accompanying GUI at https://hatmepp.warwick.ac.uk/fitting/v2/.

**Prioritising health zones.** Decision-making for gHAT strategy is challenging; national programmes have the flexibility to implement nuanced, spatially-heterogeneous interventions, however, they must adhere to more general WHO recommendations and local budget constraints. In this study, we rank strategies by how ambitious the use of additional interventions is and examine the minimum required to meet the 2030 EOT goal in each health zone—referred to here as the "preferred strategy". Maps showing the preferred strategy under different levels of certainty in EOT as predicted by the model are given in Fig. 4. Under the criterion of PEOT > 0.9 (left map), preferred strategies are defined as the strategies which achieve EOT by 2030 in at least 90% of simulations. The criterion of PEOT = 1 (right map) further restricts preferred strategies to achieve the goal by 2030 in all simulations. According to the ordered ranking (MeanAS, 40% AS, MeanAS+VC, and 40%AS+VC), the least ambitious strategy among all that meet the PEOT criterion is selected as the preferred strategy. This order of ambition ranking was based on the following principles: MeanAS represents the continuation of current intervention, 40%AS is a higher but likely achievable level of intervention, and VC is a new intervention to all health zones except Yasa Bonga. Notably, 40%AS+VC is absent in any of the preferred strategy maps because all health zones are expected to achieve the EOT goal by 2030 under the MeanAS+VC strategy which requires less resources.

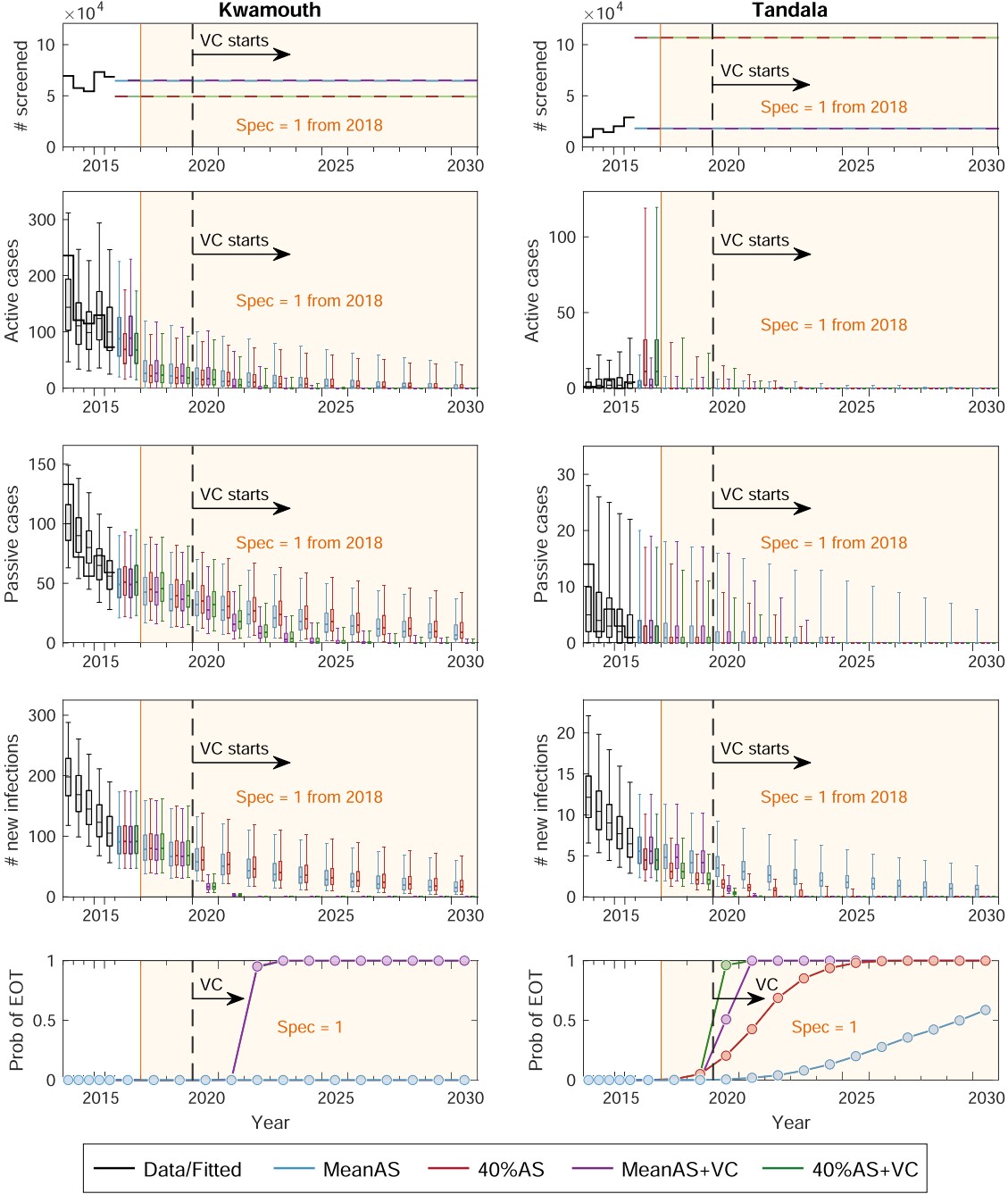

**Fig. 1 Time series of key model outputs in two example health zones.** Kwamouth (left panels) in the former Bandundu province and Tandala (right panels) in the former Equateur province represent a high-risk and a low-risk health zone, respectively. There are $n = 10,000$ independent samples, 10 from each of 1000 independent samples from the joint posterior distributions of the fitted model parameters. The top row shows the number of people actively screened, the middle three show direct model outputs (active cases, passive cases and underlying new infections from top to bottom), and the bottom row shows the probability of achieving EOT by year. Shaded regions denote that the specificity in AS was improved to 100% from 2018 in both health zones. Black lines and box plots indicate data and model fit in the last 5 years (2012–2016), coloured dashed lines denote the assumed AS starting in 2017, and colour box plots and circles present the predictions for four strategies (as defined in Table 1). Box plots summarise parameter and observational uncertainty. The lines in the boxes present the medians of predicted results. The lower and upper bounds of the boxes indicate 25th and 75th percentiles. The minimum and maximum values are 2.5th and 97.5th percentiles and therefore whiskers cover 95% prediction intervals. Full-model outputs (2000–2050) of all 168 analysed health zones are available in the graphical user interface at https://hatmepp.warwick.ac.uk/projections/v2/. AS active screening, VC vector control, Spec specificity, EOT elimination of transmission.

Maps showing lower PEOT thresholds (from PEOT = 0.5 upwards) can be found in the GUI at https://hatmepp.warwick.ac.uk/projections/v2/ and far less intensification would be required if a 50% probability of meeting the goal by 2030 is considered to be sufficient.

Switching to intensified strategies is generally expected to reduce disease burden and increase confidence that EOT will be achieved. Despite our prediction that many deaths should be prevented by either the 40%AS or MeanAS+VC strategies (Supplementary Fig. 4), the model predicts that a relatively low percentage of health zones

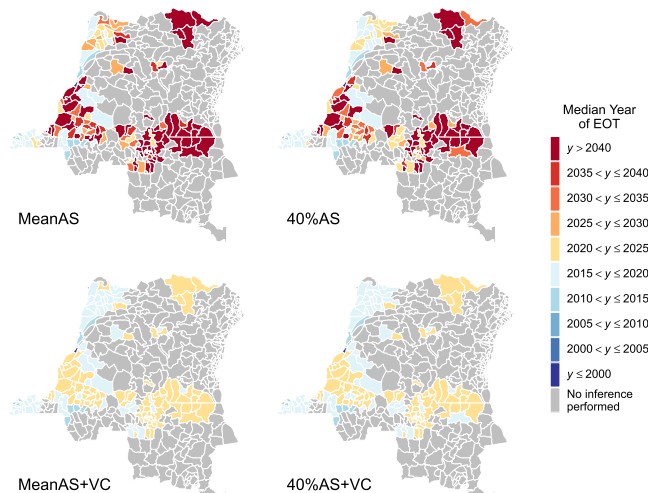

**Fig. 2 Health zone median year of elimination of transmission (YEOT) maps for the DRC.** The median YEOT provides the year in which 50% of model simulations reach EOT in each health zone. The top two maps show strategies without VC (except for Yasa Bonga health zone which is shown with VC in all maps) and the bottom maps have VC strategies with 80% vector reduction. The left maps simulate the continuation of the mean AS coverage and the right two simulate 40% AS coverage. The uncertainty of YEOT is not shown in these maps (only the average prediction). The exact median values and 95% prediction intervals for YEOT are available in the graphical user interface at https://hatmepp.warwick.ac.uk/projections/v2/. Shapefiles used to produce maps are available under an ODC-ODbL licence at https://data.humdata.org/dataset/drc-health-data. AS active screening, VC vector control, EOT elimination of transmission, YEOT year of elimination of transmission.

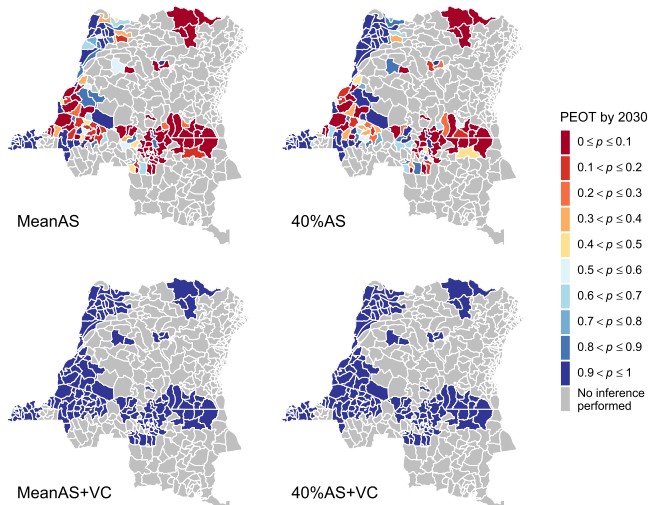

**Fig. 3 Health zone probability of elimination of transmission (PEOT) by 2030 maps for the DRC.** PEOT reveals the uncertainty of model predictions about whether EOT will occur. Health zones with PEOT > 0.9 (dark blue) will be very likely to achieve EOT by 2030, and PEOT < 0.1 (dark red) will be very unlikely to meet it. Health zones with mid-range PEOT (0.3–0.7) have high uncertainty in the success or failure of the strategy to meet the goal either because (1) the median YEOT is close to 2030, or (2) the wide distribution in the predicted YEOT. The two identical maps (with PEOT = 1 everywhere) at the bottom show that VC is an efficient tool that ensures EOT has extremely high certainty. Maps of PEOT by other years are available in the graphical user interface at https://hatmepp.warwick.ac.uk/projections/v2/. Shapefiles used to produce maps are available under an ODC-ODbL licence at https://data.humdata.org/dataset/drc-health-data. AS active screening, VC vector control, EOT elimination of transmission, PEOT probability of elimination of transmission, YEOT year of elimination of transmission.

will eliminate transmission by medical-only strategies with high probability (38% under PEOT > 0.9 and 19% under PEOT = 1). We used the historical data in conjunction with model assumptions to understand the causes of the apparent high need for VC and suggest where and what kind of intensified interventions could result in the achievement of EOT by 2030.

Using the WHO's risk categories, health zones can be classified as: moderate- or high-risk (≥1 case per year on average per 10,000), or low- or very low-risk (≥1 but <100 cases per year on average per 1,000,000). Based on data from 2012 to 2016, there are 125 health zones in low- or very low-risk categories. One may assume these health zones should be on track to meet EOT by 2030 since they have low reported cases in recent years, however, the model predicts the majority (68 health zones) need VC to achieve EOT by 2030 with more than 95% probability. The discrepancy arises from uncertainty in model predictions due to the recent low coverage of AS. It is AS which provides information on quantifying the underlying transmission and affects model predictions. In order to maximise resource efficiency, reductions in AS commonly happen when fewer cases are reported. More than 95% of the low- or very low-risk health zones screened a total of less than 50% of its population in the last 5 years (i.e. less than 10% annually). As a result, VC is favoured in model predictions and may be unnecessary in practice in low- or very low-risk health zones. For moderate- or high-risk health zones, the model predicts nearly all health zones (36 out of 43) need VC to meet EOT by 2030 with more than 95% probability. Although VC seems a reasonable tool in moderate- or high-risk health zones, unfortunately it is unlikely to be practical to roll out large-scale VC in all of them in this short timeframe.

The symptoms of gHAT are generally mild and non-specific before trypanosomes cross the blood-brain barrier, however,

progression to more severe symptoms and then death is the outcome when infected humans are missed by AS and never identified in PS. Thus, the disability-adjusted life years (DALYs) of gHAT, a broad measure of overall disease burden, are mainly comprised of deaths outside the health care system. Health zone maps of total deaths under the MeanAS strategy (Supplementary Fig. 3) and deaths averted under intensified interventions (Supplementary Fig. 4) are available in Supplementary Note 2: Results. By identifying health zones that have greater than 50 total deaths predicted in 2017–2030 under the 40%AS strategy we compiled a priority shortlist of health zones in the former Bandundu province where VC implementation is practically feasible and highly recommended by mathematical modelling: Kwamouth, Masi Manimba, Bokoro, Bagata, Mushie, Kimputu, Mokala, Bulungu, Nioki and Kenge (as shown in Fig. 5). More than 95% of health zones have mean AS coverage lower than 25%, therefore a secondary suggestion is to increase the coverage of AS, especially in the moderate- or high-risk health zones.

**Graphical user interface (GUI).** A graphical user interface (GUI), hosted at https://hatmepp.warwick.ac.uk/projections/v2/, was built to provide interactive visualisation of the data and model outputs of all 168 analysed health zones under all simulated strategies. The time series figures including the number of people actively screened, active cases, passive cases, underlying new infections and deaths are re-generated automatically when a health zone is selected from the drop-down menu or by clicking on the country map. The predicted elimination map shows a graphical summary of YEOT and PEOT. The exact median values and 95% prediction intervals for YEOT are available when

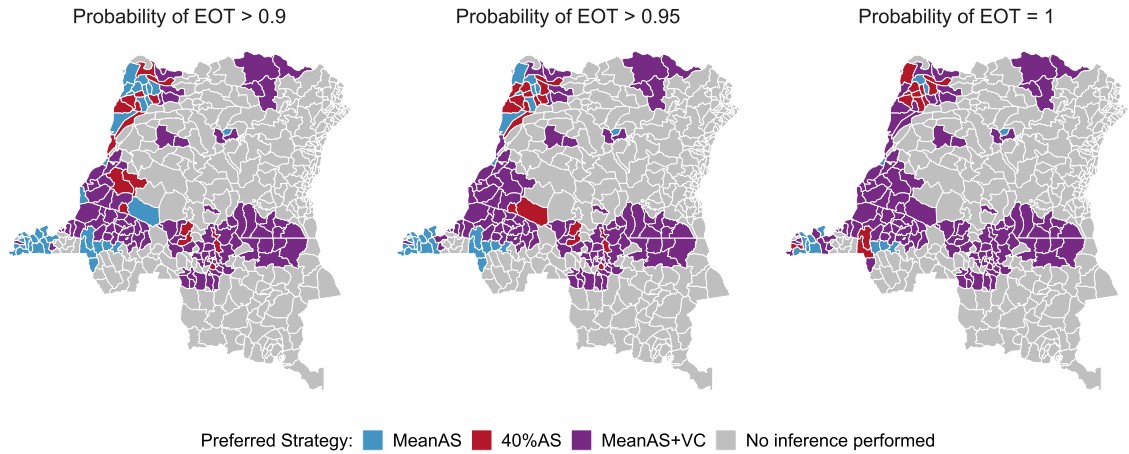

**Fig. 4 Health zone preferred strategy maps for the elimination of transmission (EOT) by 2030 in the DRC.** The preferred strategy is defined as the least ambitious strategy which is predicted to achieve EOT by 2030 with a prescribed confidence level (90%, 95% and 100%). The order of ambition ranking is MeanAS, 40%AS, MeanAS+VC and 40%AS+VC. All health zones are predicted to achieve EOT by 2030 (PEOT = 1) under the MeanAS+VC strategy so 40%AS+VC is absent here. The MeanAS and 40%AS strategies were not considered in Yasa Bonga because VC started in mid-2015. Preferred strategy maps for smaller PEOT thresholds are available in the graphical user interface at https://hatmepp.warwick.ac.uk/projections/v2/. Shapefiles used to produce maps are available under an ODC-ODbL licence at https://data.humdata.org/dataset/drc-health-data. AS active screening, VC vector control, EOT elimination of transmission, PEOT probability of elimination of transmission.

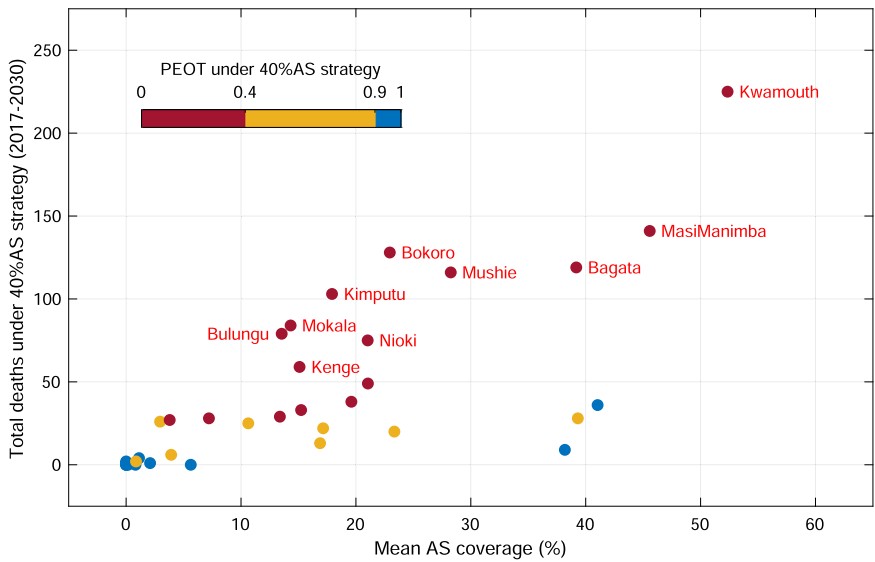

**Fig. 5 Identified health zones for supplementary vector control (VC) implementation in the former Bandundu province.** Total deaths under the 40%AS strategy present the disease burden under uniformly good AS coverage across all health zones. Colours indicate the probability of EOT by 2030 under the 40%AS strategy; blue denotes health zones that are likely to meet EOT by 2030, yellow denotes health zones with high uncertainty of achieving EOT and red denotes health zones with low chance of meeting the 2030 goal. Our model identifies a priority list of ten health zones (highlighted by their names, Kwamouth, Masi Manimba, Bokoro, Bagata, Mushie, Kimputu, Mokala, Bulungu, Nioki and Kenge) for consideration for supplementary VC implementation because these health zones have high disease burden and low chance of achieving EOT under good AS coverage. Health zones outside of the former Bandundu province are excluded from our modelling recommendation because of the concern of implementing feasibility. AS active screening, VC vector control, EOT elimination of transmission, PEOT probability of elimination of transmission.

hovering over health zones on the map. While the map defaults to PEOT by 2030, maps of PEOT by other years (2020–2040) are available via a controlling slider. In a separate tab, the preferred strategy map is shown and defaults to a threshold of PEOT > 0.9. Preferred strategy maps for lower PEOT thresholds are available by selecting the desired PEOT level via the controlling slider.

## Discussion
The presented model predictions are based on the previously fitted deterministic mechanistic gHAT model (to 2000–2016 case data)[30] and assumptions of different strategies starting from 2017. Using a mechanistic, SEIR-type model not only provides parameter estimates that describe the dynamics of infection but also allows us to simulate future dynamics under various strategies, including those not conducted in the health zone previously. Although stochastic models naturally capture the stochasticity of chance events when approaching EOT, similar dynamics and predicted EOT years were reported at the health zone level from both the stochastic and the deterministic gHAT models[31–33]. Other simpler model variants published elsewhere are not considered in this study as none of them can achieve good health zone-level fits to the WHO HAT Atlas data aggregated by health zone and year[22,25,29].

This is the first publicly available analysis of gHAT predictions for health zones across the whole of the DRC, and highlights regions we expect to be successful, and those where there may be challenges in managing disease burden and achieving the WHO 2030 target of EOT. Our custom-built GUI provides an interactive, user-friendly way to visualise these results and recommendations. By providing estimated deaths (a broad measure of overall disease burden), average predictions and 95% prediction intervals for when we expect EOT to be met, and also the probability of meeting the goal by 2030 in each health zone, we aim to quantify not only regions which may need intensified strategies, but those where current data may not be sufficient to generate predictions with high certainty.

The preferred strategy maps in Fig. 4 show that the MeanAS +VC strategy (or another intensified strategy) is likely needed in a large proportion of the health zones. This finding brings up a serious concern about the feasibility of scaling up VC in order to achieve EOT in 10 years under resource constraints. The implementation of VC began in the southern part of Yasa Bonga in 2015 and expanded to cover three large rivers (Lukula, Kafi, Inzia) and selected tributaries linked to fish ponds by 2017[34]. Scaling up of VC was slow and its feasibility was mainly determined by the availability of financial and human resources for this new intervention. In the present study, the integration of data, model assumptions, and model predictions identifies a priority shortlist of ten health zones with high gHAT burden: Kwamouth, Masi Manimba, Bokoro, Bagata, Mushie, Kimputu, Mokala, Bulungu, Nioki and Kenge as regions where VC is predicted to be a beneficial supplementary tool – all these health zones are predicted to have > 50 gHAT deaths between 2017 and 2030 and also unlikely to meet the EOT target by 2030 under the 40%AS strategy. Comparing our priority list for VC to the planned VC scale-up in the DRC guided by recent case data, modelling and habitat suitability (https://www.lstmed.ac.uk/projects/tryp-elim-bandundu), six of ten priority health zones identified by our model are currently targeted as operational areas for VC expansion. Furthermore, our model suggests that the four other health zones earmarked for scale-up (Bandundu, Kikongo, Bolobo and Yumbi) would have been expected to have ≥25 deaths without VC interventions.

Other health zones predicted to miss the 2030 EOT goal could also benefit from this tool, although careful consideration is required to assess whether scaling up medical interventions is easier to implement than introducing large-scale VC. The reported effectiveness of VC is high in general but the variations between locations are non-negligible. According to our sensitivity analysis on the effectiveness of VC (Supplementary Fig. 2), the time difference in achieving EOT could be several years longer with only 60% annual tsetse reduction, but this is still substantially faster than with medical-only interventions in many settings. Our model forecasting would be more accurate if the location-specific effectiveness of VC—which remains unknown in most health zones—was taken into account. Pessimistic model predictions can be found in some health zones where the coverage of AS is very low recently or historically. Low AS coverage creates additional uncertainty in model outputs and therefore can make model predictions overly pessimistic (i.e. they could overstate the need for VC in low- or very low-risk health zones). Exploring the minimum AS coverage required to achieve EOT by 2030 would be another mathematical modelling approach to address where and what kind of intensified interventions are needed to achieve EOT.

Whilst our policy recommendations are based on the expected infection, disease burden and reporting dynamics in the next decade, we also report long-term projections up to 2050. Making long-term predictions is always a challenge for epidemiological modelling in terms of uncertainty in a range of factors (e.g. demographic and environmental changes). However, presenting simulations up to 2050 provides illustrative outcomes beyond 2030, such as whether we expect the EOT goal to be missed by a few years or if it might be totally infeasible. Whether elimination is expected to occur before 2030, slightly beyond 2030, or after 2040 is a metric of value which our projections can be used to assess. Our projections are used to consider whether current strategy is on the right track to meet policy objectives or whether a change in strategy is warranted; high levels of accuracy in model projection at 2050 are not necessary for modelling to prove useful in these assessments.

When the new data from 2017 onward becomes available, we will be able to use it to validate our model by changing assumed AS coverage to actual numbers of people screened and then comparing the predicted active and passive cases to reported cases. Subsequent re-fitting to the recent case data would further refine model predictions presented here and is an important step in the continuous process of modelling to support policy under NTD-PRIME principles[35] (Supplementary Table 3). Our model framework is flexible and could be used to predict the impact of unexpected future changes by estimating how they could alter observable (i.e. reported cases and deaths) and unobservable variables (i.e. new infections); in the present climate of the COVID-19 pandemic and recent Ebola outbreaks in gHAT-endemic parts of the DRC, this is particularly relevant and could provide support in planning whether subsequent gHAT interventions should be altered due to unforeseen interruptions. Other work by this modelling group specifically analysed the potential impact of different COVID-19 interruptions to the gHAT programme in the DRC and concluded that short interruptions to active screening coverage only were unlikely to result in delays to EOT due to the long timescale of disease progression in humans[33]. If longer interruptions occurred and impacted both active and passive screening then we would expect delays of similar duration to the interruption itself, however, it is now known that this worst-case scenario did not take place and the main impact on the DRC programme was lower coverage of active screening during 2020.

Without taking into account the potential impacts of animal reservoirs, asymptomatic infections and secondary infections from host/vector movement, our model results could be optimistic on the issue of gHAT burden and EOT. However, our predicted regions for enhanced control will remain on the list should these alternative model formulations be utilised in future modelling exercises. The impact of other factors such as the screening of high-risk populations and the presence of animal reservoirs on gHAT transmission have been studied by mathematical modelling[22,25,26,29,36,37]. Recruiting high-risk individuals can, unsurprisingly, improve the effectiveness of AS and bring down the YEOT substantially[26]; the present framework could be extended to quantify the impact of this type of improved AS. Models considering an animal reservoir have largely been inconclusive about the presence of zoonotic transmission (when fitted to longitudinal human case data) however they have indicated that animal reservoirs are unlikely to maintain the infection by themselves[22,25]. An analysis including animal reservoirs could yield different results for estimated deaths and YEOT predictions presented here, although our previous work suggests that we would probably not expect large qualitative differences. Another concern is that transmission could be maintained through asymptomatic humans[38,39]. Although a few modelling studies have utilised frameworks explicitly incorporating asymptomatic human infections[29,37,39], there is limited observational data to parametrise them with high certainty, and it is unclear how their inclusion in this study would impact projections. Host or vector

movement may play some role in spreading infection. Health zones that achieve EOT earlier but are neighbouring higher prevalence locations have a higher chance of resurgence due to human mobility. The typical size of health zones in the DRC is much bigger than the spatial scale of tsetse movement, although epidemiological foci may overlap health zone boundaries and could impact predictions. A stochastic model applied at a smaller spatial scale is more suitable for addressing issues related to chance events and their impact on EOT[28]. Given that our predictions are possibly optimistic, yet we still find health zones reporting cases and unlikely to meet the 2030 EOT goal, this highlights the urgent need to strengthen interventions in these locations.

A new oral drug to treat gHAT—fexinidazole—has now been approved for use in the DRC, and is being utilised in the country. Despite the obvious advantages for patients, ease of transport and administration, it is not deemed suitable for use in individuals without parasitological case confirmation[40], and hence is unlikely to greatly impact on transmission as part of a strategy. A second oral drug—acoziborole—hoped to be a safe single-dose cure, is under clinical trial and could, in principle, radically change the paradigm of diagnostic and treatment algorithms, especially in an AS setting[41]. The non-toxic compound used in acoziborole may allow mobile screening teams to "overtreat" rapid diagnostic test-positive, gHAT suspects without parasitological confirmation. Another important tool to measure elimination or detect re-emergence of gHAT as prevalence approaches zero is its diagnosis. In contrast to the need to detect as many cases as possible in epidemic and endemic situations, avoiding any false-positive results becomes more important when the prevalence is low. A promising new test with high specificity and sensitivity, iELISA, was newly developed for this purpose[42]. Different from the existing trypanolysis test, iELISA has lower requirements at both the laboratory and the technical skill levels and therefore may be a better tool for post-elimination monitoring as endemic countries should be able to sustain it with little external financial and technical support. Mathematical modelling could be used to investigate the impact of potential diagnostic and treatment algorithms and predict the impact of such strategies using aco-ziborole and iELISA on EOT before they begin. These types of novel interventions could be particularly helpful as we approach the endgame for gHAT.

AS planning by PNLTHA-DRC is guided at a village level by WHO recommendations. These include screening historical gHAT prevalent villages and stopping AS after 3 years of zero case detection and then switching to "reactive AS" when new cases arise in PS[16]. Disease foci (transmission pockets) and intervention areas may only cover small parts of the health zone. We did not consider the geographical distributions of reported cases within health zones in our model. Although our model already produced fits well-matched to data, we expect that the relative risk of high-risk humans ($r$) is underestimated and the proportion of low-risk humans ($k_1$) is overestimated due to the assumption of homogeneity within health zones. Model fitting and projections at smaller scales would provide more realistic impacts on the efficacy of AS and VC when infections are highly clustered in smaller areas or when incidence becomes very low and triggers the cessation of AS. To achieve smaller-scale analyses in the future there are a range of computational challenges that would need to be overcome, such as an order of magnitude increase in the number of locations to be fitted to, and greater levels of stochastic noise in the data. In some smaller-scale regions, there may simply not be sufficient historical data points to allow for robust fitting of the model.

In this study, our four strategies were assumed to carry on indefinitely without any stopping, however, the economic gains

and health risks associated with cessation should be examined. A previous health economic analysis concluded that VC can be cost-effective at low willingness-to-pay thresholds per DALY averted in high-risk settings[43]. Taking account of the PNLTHA-DRC algorithm of reactive screening, a novel health economic analysis based on predicted model dynamics would allow for the examination of cost-effective strategies rather than the "preferred strategy" presented here based on a cruder ranking of "ambition".

Looking across at other infections targeted for elimination, the enormity of the challenge ahead becomes apparent—with many of these programmes reaching ever-lower levels of disease, but failing to meet elimination deadlines. Modelling in this study suggests that, even though the elimination of gHAT in the near future may be epidemiologically feasible with current tools, its widespread, low-level persistence across the DRC could prove operationally challenging for the achievement of the goal in the short term. In many regions, there is considerable uncertainty whether current interventions are sufficient to meet EOT in the next 10 years, yet the prospect of intensifying strategies in dozens of health zones may pose a large, possibly insurmountable, burden on both financial and personnel resources. As further progress is made towards the elimination of gHAT, it will become increasingly important to use data-driven methods to optimise the endgame pathway based on practical strategies and use these methods to quantify success.

## Methods

**Model.** Previous modelling studies explored several model variants for some health zones within the former Bandundu province in the DRC and a focus in Chad. They showed that a simpler model variant ("Model 1") which is a Ross–Macdonald style model was unable to fit the case data well in the DRC or Chad and therefore concluded that heterogeneous risk of human infections, participation in AS structures and improvement in passive screening are essential to perform a good model fit to observed longitudinal human case data[22,25,29,44]. Stochastic modelling studies on gHAT dynamics towards EOT demonstrated surprisingly similar results between deterministic and stochastic models despite wider prediction intervals for EOT years by the stochastic model[31–33]. In this paper, we considered a passive improvement on top of a previously developed variant ("Model 4") of the Warwick deterministic gHAT model[25,26,30,44], which captures systematic non-participation of high-risk groups in the population— anecdotally believed to be working-age people spending time near tsetse habitat, and away from villages during active screening activities, to predict gHAT dynamics by considering transmission among humans, tsetse and non-reservoir animals.

As illustrated in Fig. 6, human hosts' risk of infection categories are defined by their different contact rates with tsetse. High-risk humans (subscript $H4$) represent the working-age males and are $r$-fold more likely to receive bites than low-risk humans (subscript $H1$). Any blood meals taken upon "other" hosts do not result in infection. Both the proportion of low-risk humans ($k_1$ from which we get the proportion of high-risk humans, $k_4 = 1 − k_1$) and the relative bites on high-risk humans ($r$) are fitted parameters in our model because we believe they would vary geographically. Tsetse select their blood meal from one of the host types dependant upon innate feeding preference and relative host abundance. We assume tsetse preferentially feed on humans with a probability $f_H$ which is taken to be 0.09[45], if some other fixed value of $f_H$ was used this would impact the fitting of the other model parameters, in particular $m_{eff}$ in Supplementary Eq. (1). In contrast to the assumption that low-risk humans randomly participate in active screening, high-risk humans are assumed to never participate. This participating structure is supported by data in previous model fits[30]. All infected individuals are assumed to exhibit treatment-seeking behaviour regardless of risk, however, those in early-stage infection have a much lower probability of seeking treatment compared to those with late-stage infection. Thus passive detection is assumed to be dependent on their disease progression (slower rate to detection for stage 1, $\eta_H$ compared to stage 2, $u\gamma_H$), which includes the health zone-specific availability of fixed health facilities with gHAT diagnostics and underreporting. Tsetse bites are assumed to be taken on humans or non-reservoir animals. However, the non-reservoir animal species do not need to be explicitly modelled, i.e. this model variant does not include tsetse to non-human animal transmission. Complete mathematical descriptions are available in Supplementary Methods with detailed updates in Supplementary Note 1: Model Updates.

**Fitting.** In a previous publication, we used the gHAT case data and annual screening numbers in 168 health zones from the WHO HAT Atlas to individually fit the gHAT model to health zone-specific trends from 2000 to 2016[30]. The data are aggregated by location, year, surveillance type and diagnosed disease stages (stage 1, stage 2 and stage

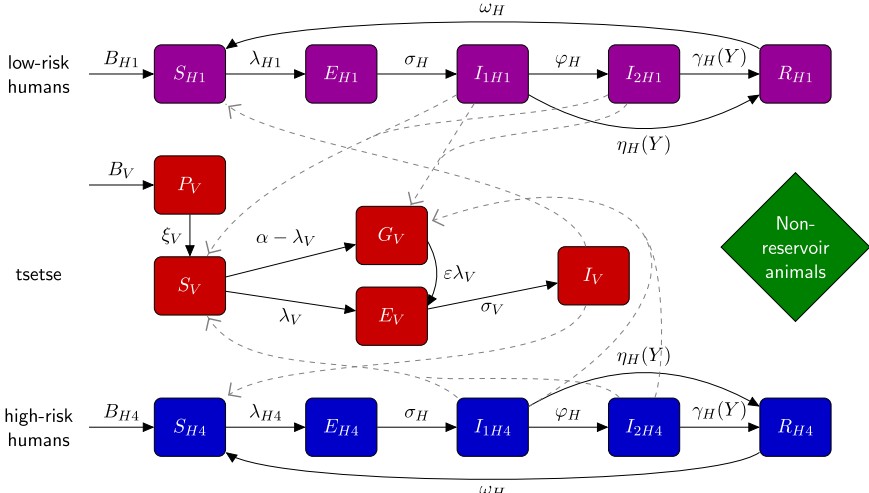

**Fig. 6 Illustration of compartmental gHAT model.** The multi-host gHAT model is composed of one host species able to confer gHAT (humans), a further non-reservoir species (others) and tsetse. After the incubation period, infected human hosts follow the progression which includes infectious stage 1 disease, $I_{1H}$, infectious stage 2 disease, $I_{2H}$, and non-infectious (due to hospitalisation) disease, $R$. Pupal stage tsetse, $P_V$, emerge into unfed adults. Unfed tsetse are susceptible, $S_V$, and following a blood meal become either exposed, $E_V$, or have reduced susceptibility to the trypanosomes, $G_V$. Tsetse select their blood meal from one of the host types dependant upon innate feeding preference and relative host abundance. High-risk humans are more likely to receive bites than low-risk humans. Any blood meals taken upon "other" hosts do not result in infection. The transmission of infection between humans and tsetse is shown by grey paths. This figure is adapted from the original model schematic[25], which was published under a CC-BY licence.

unknown). Location was defined by the available geolocation and geographical identifier information, while surveillance type was either active or passive screening[12,46,47]. Cases of gHAT are not reported across all of the DRC (only 290 out of 516 health zones had any data for the 2000–2016 period), and some health zones have very little case reporting or screening activities (122 health zones of the 290 reporting any data). We do not include these locations in either fitting to the historical data or future predictions and they are coloured grey on corresponding maps. Whilst we are not willing to make quantitative predictions for the health zones coloured grey on our maps using our model, we believe that these largely represent regions without transmission. For each health zone we fitted, we estimated model parameters that are likely to geographically vary across regions including basic reproduction number, the proportion of low-risk humans, relative bites taken on high-risk humans, and reporting bias such as passive treatment rates for both stages, passive reporting proportion and active screening diagnostic specificity (Supplementary Table 2). The fitted model also takes into account previous advances in medical, diagnostic, and control systems and assumes all false positives found in active screening are assigned to stage 1. Samples from posterior distributions of parameters were obtained by fitting to annual health zone-level data for the period 2000–2016 using an adaptive Metropolis-Hastings MCMC method[30]. Posteriors and fitted trends to case data are available at https://hatmepp.warwick.ac.uk/fitting/v2/.

**Forward projections**. Major changes during the data collection period include improvements to the PS systems in the former provinces of Bandundu and Bas Congo, improved active case confirmation via a video recording of diagnostics in Mosango and Yasa Bonga in the former Bandundu province from 2015, and implementation of large-scale VC in Yasa Bonga since mid-2015. Based on the continuation of the current PS system, we considered four strategies for projections from 2017 to 2050, which included different coverage of AS and whether or not to implement VC from 2020. As summarised in Table 1, the level of AS in a health zone is assumed to be either at the recent (2012–2016) observed mean or at 40% of its population, and hence depends on the historical data and population size in each health zone. HistMaxAS and HistMaxAS +VC strategies (historical maximum level ever achieved in AS) were included in our projections in Supplementary Note 2: Results. For VC, a fixed effectiveness of 80% tsetse reduction after 1 year was used in the strategies with VC in all health zones except Yasa Bonga, where effectiveness of 90% has been reported[34]. Other tsetse reductions (i.e. 60% and 90%) were considered in sensitivity analyses in Supplementary Fig. 2. Further model assumptions include: (1) in Yasa Bonga only strategies with VC are considered since VC was already in place before 2017; (2) video confirmation of parasitological diagnosis was included from 2018 in Bandundu health zones to avoid false-positive diagnoses in AS; (3) automatic improvement of the diagnostic algorithms to 100% specificity outside Bandundu when the detected case numbers were close to the expected incidence of false-positive detections.

The dataset finished in 2016 and so forward projections were performed from 2017 to 2050, independently for each health zone. Parameter uncertainty was represented by 1000 randomly selected sets of parameters from the health zone-specific posterior distributions from the model fitting. Observational uncertainty in predicted case numbers each year was considered by drawing ten random samples from the predicted mean dynamics for each set of parameters. In model outputs, 10,000 samples for observable variables such as active and passive cases, and related metrics were generated. On the other hand, unobservable variables like new infections and the year of EOT were predicted by the 1000 model realisations (parameter uncertainty but no sampling uncertainty).

**Measuring elimination of transmission**. As there is no direct way to observe EOT, the WHO suggests a primary indicator of zero reported cases to measure the achievement of EOT[13,48]. However, the number of reported cases depends largely on the strength of medical interventions, so other methods to assess progress towards EOT are desirable to complement imperfect case indicators[47,49].

Fortunately, mechanistic modelling provides the means to both infer and predict the unobservable transmission dynamics to assess EOT. Here, we calculated the number of underlying new infections each year in the epidemiological model. Unlike the discrete nature of populations, the outputs of deterministic models are continuous and whilst they can asymptote to zero they will never reach it. Therefore, to identify a realistic point at which EOT has been achieved, we introduced a proxy threshold (=1) for annual new infections and assume that EOT is achieved when the number of new infections is below the threshold (Supplementary Note 1: Model Updates).

**Reporting summary**. Further information on research design is available in the Nature Research Reporting Summary linked to this article.

## Data availability
Case data used for model fitting in Crump et al.[30] and screening data used to inform future potential screening coverage were obtained through the World Health Organisation (WHO) HAT Atlas. Identifiable data cannot be shared publicly because of our data-sharing agreement with the WHO's HAT Atlas which is under the stewardship of the WHO. Data are available from the WHO (contact neglected.diseases@who.int or visit https://www.who.int/trypanosomiasis_african/country/foci_AFRO/en/) for researchers who meet the criteria for accessing confidential data. Timeframe for response to requests would depend on the WHOs schedule.

## Code availability
The Matlab code used to simulate this work is available in Open Science Framework at https://osf.io/jza27/.

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

## Acknowledgements

The authors thank PNLTHA for original data collection, WHO for the data access (in the framework of the WHO HAT Atlas[48]), and Cyrus Sinai and Nicole Hoff from UCLA Fielding School of Public Health for providing health zone-level shapefiles (current versions can be found at https://data.humdata.org/dataset/drc-health-data). This work was supported, in whole or in part, by the Bill & Melinda Gates Foundation [OPP1177824 (C.H., R.E.C., P.B., K.S.R. and M.J.K.), OPP1184344 (K.S.R., S.E.F.S. and M.J.K.), OPP1156227 (K.S.R., S.E.F.S. and M.J.K.), OPP1186851 (K.S.R., S.E.F.S. and

M.J.K.), OPP1155293 (E.M.M. and C.S.)]. Under the grant conditions of the Foundation, a Creative Commons Attribution 4.0 Generic License has already been assigned to the Author Accepted Manuscript version that might arise from this submission. This work was also supported by the Directorate-general Development Cooperation and Humanitarian Aid (E.M.M. and C.S.). The funders had no role in study design, data collection and analysis, decision to publish or preparation of the manuscript.

## Author contributions

C.H., R.E.C. and K.S.R. developed the software and performed the analyses. R.E.C., C.H. and P.B. visualised the results. R.E.C., E.M.M. and C.S. analysed the data. K.S.R. developed the methods. C.H. and K.S.R. wrote the original draft. K.S.R., S.E.F.S. and M.J.K. conceptualised the study. All authors reviewed and approved the final version for publication.

## Competing interests

The authors declare no competing interests.
