## [Peer Review File · Nature Communications]

Identifying regions for enhanced control of *gambiense* sleeping sickness in the Democratic Republic of CongoREVIEWER COMMENTS

Reviewer #1 (Remarks to the Author):

In this manuscript, Huang and colleagues performed a mathematical modeling study to explore the capacity to achieve the elimination of transmission (EOF) of gHAT at the year 2030 in DRC. The authors mainly used their published model and likelihood method to estimate disease parameters and predict disease trend in the following years. In my opinion, this work is more suitable for a specific health-related journals (e.g., Bulletin of the WHO, IJE, or BMC Medicine). Considering the Aims and Scope of the Nature Communications (i.e., Papers published by the journal aim to represent important advances of significance to specialists within each field), the expected publications may be better to at least have some improvements in the methodology and provide more insights that can guide other experts how to improve the specific fields. However, I don't think there is sufficient contributions along this direction.

My detailed comments are summarized as below:

(1) If I understood correctly, the authors used a deterministic model to fit the case-based incidence data. The authors may need to explore the influence of stochasticity. It will be valuable if the authors could check results with stochastic model.

(2) It's unclear how the authors fit the model to the data.

(3) There is no model comparison. It's unclear whether simpler models can work better. For example, complex models may lead to overfitting. So it's unclear whether the projection is robust or not.

(4) There is no discussion about the identifiability of model parameters. For example, the authors may need to generate some simulation data, and see if the inference method can work well to identify the true parameters. Otherwise, it's hard to see whether the inference and projection were done correctly.

(5) It's unclear whether the authors considered the reporting bias to detect cases. For example, different locations in DRC may have different reporting rates. So that some parameters related to the reporting system are needed.

(6) The model structure is simply considered as a single population for each health zone in DRC. It's not clear whether human mobility and the movement of the vector will affect the distribution of cases.

(7) Are you fitting each location separately or fit all locations together using a single likelihood?

Reviewer #2 (Remarks to the Author):

Comments are in the attached file

This is an interesting and useful paper that uses an existing and well published model framework at a national scale to try to give information about how to prioritise HAT control resources.

For context I have been conducting the review of this paper whilst considering possible areas outside former Bandundu that could benefit from vector control, so being able to review this paper has been very useful, but has also raised a lot of questions about the model.

At a high level I have misgivings about presenting the results purely in terms of time to elimination of transmission. I think that a broad piece such as this going into a high impact journal the results should be primarily presented in terms of public health impact with elimination of transmission as a secondary output. Elimination of transmission is a target set out by funders and stakeholders, but is merely a concept to the teams implementing controls on the ground who measure impact in terms of public health impact. “Why are you proposing we work in this health zone?” ANS “Because the model says that it will not achieve elimination of transmission before 2040” is quite a weak response. “Why should we work in this health zone” ANS “Because it will prevent XXX deaths” carries a bit more punch. Elimination of transmission is certainly important but must be considered alongside the public health impact.

Also at a high level. You don’t make much of it, but your results give some strong indications about the efficacy of active screening to drive elimination. I think that people will be surprised by how poor are the positive predictive values of active screening detections and these are only going to get worse. This poor PPV is part of the reason why active screening is predicted to play relatively little part in the targets. It is a surprising result and as a consequence needs more attention – more below.

In terms of deciding where it may be possible to site vector control activities one’s eye is drawn on your maps to the massive clumps of dark red in Maniema and Katanga. One’s eye is drawn here because they are not areas that have a high HAT burden and so were not on the radar for many. Yet they are predicted to fail to make the elimination targets which some higher incidence ZS are predicted to hit. Because it is surprising and unexpected does not make it wrong but does require additional scrutiny – one needs to be very sure before directing resources to low-burden areas.

Major corrections

1. Please make the results clearly available in terms of public health impact (this could be as DALYs, mortalities averted, infections averted or YLLs). I think that this is useful for people on the ground and those planning activities, particularly as from your model disease burden and time to elimination are not well correlated and we don’t want people to conflate them
2. meanAS and maxAS. These are based on historical precedent, but can often vary 10x between health zones. meanAS I don’t have a big concern with because it is showing the business as usual scenario. maxAS is a problem. Unless the reader really digs they could be forgiven for assuming that maxAS represents the same screening effort across all health zones, not potentially a 10x difference. Furthermore, it is really difficult to find out what maxAS represents in terms of screening effort. Why when VC is based on a theoretical target can we not have a theoretical target for AS to compare between health zones. Why would it be so difficult for a ZS to raise its

screening from a previous peak of 5% to 10%, particularly when one considers that active screening can be scaled up or redeployed much faster than vector control resources. Hence, when I look at these ZS in Maniema and Katanga I wonder whether they are a particular problem area for HAT that really requires vector control or whether they are just areas that have never received much active screening

3. Homogenous risk. Your model assumes that the risk and incidence is spatially homogenous within the ZS, albeit with high risk individuals that are not actively screened. This seems fine in the higher incidence health zones in which generally the majority of the population is at risk (albeit likely with local variations in the magnitude of the risk). My concern is with the lower incidence health zones where you have had to assume that the low incidence is homogeneously smeared across the ZS and so the whole population is at low or very low risk. In some ZS this may be the case, but in reality it is more likely to be highly focalised with areas of high risk populations and larger portions of the population at very low or negligible risk. Where I think that this matters is that it contributes to the low efficacy of AS. If maxAS is 5% in one low incidence ZS and 50% in a higher incidence ZS then you are modelling that translates to 5% and 50% of the populations at risk getting screened. However, active screening is targeted to those villages that are perceived to be at risk (or at least have a recent history of HAT) – your co-authors from PNLTHA could expand on this. So, considering this, if in our low incidence ZS only 10% of the population is at risk then the screening will be targeted at this population, so the effective screening rate would be 50%.
4. (I think) Your estimate of specificity is based on an estimate of the proportion of *all* cases that are false positives. This may not impact the results or the estimates of specificity, but to do this correctly I think that you need to separate HAT cases by stage. HAT cases in stage 2 are much less likely than cases in stage 1 to be false positives. This is because to be a stage 2 HAT cases there must be at least 2 of 4 or 5 positive indicators. To be stage 2 a case must have ((positive parasitology and / or strong clinical signs and / or CATT dilutions) AND (trypanosomes in CSF and / OR elevated WBC count)). It is not without error, but much less likely to produce false positives than stage 1 which requires just one indicator. In my view specificity would be better fitted using just stage 1 cases and assuming that all stage 2 cases all true positives.
5. You need to present the fitted parameters for all ZS in supplementary information or through the GUI. If we are thinking about these timelines to elimination as a linear equation then elimination year (y) is $y = a + bx$ where a is the number of HAT cases at the start of predictions x is time and b is a function of HAT control and the fitted epidemiological parameters. So, to use an example we can compare Samba ZS in Maniema to Bagata ZS in Bandundu. Our value for a in Samba is 27 but it is 47 in Bagata. However, under the basic scenario y is 2039 for Bagata but > 2040 for Samba. This must mean that the differences lie within b . Now it could be down to the meanAS parameter, or it could also be within the fitted parameters. Hence the reader needs to know where the cause lies.

Minor comments.

Introduction

Line 3. The word "yet" is erroneous

Line 4. I would remove the parenthesis with (potentially illusive), either way, I think you mean elusive

Lines 2-7 (particularly 5-7 really require some evidence to support them

Lines 13-16. A long and clunky sentence. Give the reader a hand here

Line 21. English language pedantry. Replace "most" with "greatest number of..."

Line 24. Is it? Where it will end up depends upon the direction of travel as the starting point, so without a reference this is a weak statement, particularly in the context of a disease which has little control in some countries.

Line 26. "has" not "have" and can you give us more information here on the distribution of populations of HZs. As worded, this suggests that they are quite uniform in size which I didn't understand to be the case, but may bely my ignorance. You should also add some information for the reader that public health care is managed at the HZ level

Line 32. How are the 168 HZs defined as endemic?

Introduction generally. Is very light indeed on references 13 in total of which 7 are references to your own group's modelling work. There are a lot of big statements without any support and a lot of HAT-specific terms and concepts which the wider reader may not be familiar with (such as VC and AS) which the reader may not be familiar with in the HAT context and gets no referential support.

Methods (because methods logically come after the introduction)

Lines 226-231. See my general comment on heterogeneity

Line 255. Add a space after future

Table S2. Are the parameters below d_{change} supposed to be there?

Results

Figure 1 (& GUI). Please mark the point at which AS specificity goes up to 100%

REVIEWER COMMENTS

Reviewer #1 (Remarks to the Author):

“In this manuscript, Huang and colleagues performed a mathematical modeling study to explore the capacity to achieve the elimination of transmission (EOT) of gHAT at the year 2030 in DRC. The authors mainly used their published model and likelihood method to estimate disease parameters and predict disease trend in the following years. In my opinion, this work is more suitable for a specific health-related journals (e.g., Bulletin of the WHO, IJE, or BMC Medicine). Considering the Aims and Scope of the Nature Communications (i.e., Papers published by the journal aim to represent important advances of significance to specialists within each field), the expected publications may be better to at least have some improvements in the methodology and provide more insights that can guide other experts how to improve the specific fields. However, I don't think there is sufficient contributions along this direction.”

Author response:

We would like to argue that this article, on the elimination of gHAT in the DRC, is not only of interest to the readership of Nature Communications because it makes substantial advancements in the current gHAT elimination literature and is applicable to others working in infectious diseases, but also because it is of significant interest to a broad range of readers interested in public health, modelling and/or data-driven decision making.

Alongside its partner paper, (please see the preprint in MedRxiv: <https://www.medrxiv.org/content/10.1101/2020.08.25.20181982v2>) which was simultaneously submitted to Nature Communications, the two articles focus on the epidemiological and economic feasibility of gHAT elimination in the DRC, the country with the largest disease burden. Previous modelling work for the DRC has not been as expansive, nor as tailored as this current study. Previous work at a health zone level by Rock *et al* (Epidemics, 2017) focused exclusively on the former province of Equateur in the north west of the country covering only 57 health zones, compared to the 168 health zones covered in the current study, and included only 723 data points compared to 5332 as used for fitting in this study. This article therefore makes substantial advancements both in terms of the geographical coverage of predictions (tripled the geographical coverage, from 11% to 33% of the health zones) but also with respect to the tailoring of the fit for location specific predictions through the use of 17 years of 168 health-zone-level case and intervention data (13 years of 57 health-zone-level data in the previous study). The projections generated by the model and under a range of plausible strategies at a national scale are a powerful tool that can be used by the DRC national sleeping sickness programme and their partners to guide future intervention planning and policy decisions. With this in mind we have also created a graphical user interface (GUI) which displays all the modelling results from the paper in a platform that is user-friendly and interactive. Furthermore, researchers working in other infectious diseases can utilise this study as a framework to understand how to synthesise on-the-ground knowledge, programme data and modelling to produce location-specific predictions and strategy recommendations to reach specific goals. Finally, the rationale for submitting the articles as a pair in Nature Communications is due to the

fact that the results generated by this present study were used to perform a cost-effectiveness analysis to examine the efficiency of gHAT elimination strategies across different prevalence settings. We believe the breadth of information is too large for a single article. Combined, these papers highlight the real challenges faced by many infectious disease elimination/eradication programmes as cases reach ever closer to zero.

To ensure that the text sufficiently reflects the novelty and research progress generated by this current work we adjusted our text as below (line 172 and 183):

“This is the first publicly available analysis of gHAT predictions for health zones across the whole of DRC, and highlights regions we expect to be successful, and those where there may be challenges in achieving the WHO 2030 target of EOT. Our custom-built GUI provides an interactive, user-friendly way to visualise these results and recommendations.”

“The integration of data, model assumptions, and model predictions identifies a priority shortlist of ten health zones with high gHAT burden: Kwamouth, Masi Manimba, Bokoro, Bagata, Mushie, Kimputu, Mokala, Bulungu, Nioki, and Kenge as regions where VC is predicted to be a beneficial supplementary tool – all these health zones are predicted to have >50 gHAT deaths between 2017–2030 and also unlikely to meet the EOT target by 2030 under 40%AS strategy. Comparing our priority list for VC to the planned VC scale-up in DRC guided by recent case data, modelling and habitat suitability (<https://www.lstmed.ac.uk/projects/tryp-elim-bandundu>), six of ten priority health zones identified by our model are currently targeted as operational areas for VC expansion.”

Our detailed responses to specific reviewer comments are summarised below:

Comment	Author response
(1) If I understood correctly, the authors used a deterministic model to fit the case-based incidence data. The authors may need to explore the influence of stochasticity. It will be valuable if the authors could check results with stochastic model.	You are correct that the underlying model is deterministic in nature, however we do account for some stochasticity in case reporting in both active and passive screening, as well as uncertainty in parameter estimates. In other published work using similar model variants we have shown how the results of deterministic (ODE) and stochastic (tau-leaping) variants of our gHAT transmission model lead to very similar results in spite of the very low prevalence observed in many settings in DRC (e.g. Castaño et al, JID, 2019; Davis et al, CID, 2021; Aliee et al, Trans of the Royal Soc Trop Med & Hyg, 2021). We added the following at the beginning of our discussion (line 167) to address this: “Although stochastic models naturally capture

	the stochasticity of chance events when approaching EOT, similar dynamics and predicted EOT years were reported at health zone level from both the stochastic and the deterministic gHAT models [31-33].“ We also note that the main difference we expect if we were to use a stochastic transmission model would be wider prediction intervals for EOT years rather than substantial changes to average predictions. We added the following in the methods section (line 276): “Stochastic modelling studies on gHAT dynamics towards EOT demonstrated surprisingly similar results between deterministic and stochastic models despite wider prediction intervals for EOT years by the stochastic model [31-33].”
(2) It's unclear how the authors fit the model to the data.	The model was fitted to historical data using an adaptive Markov Chain Monte Carlo (MCMC) procedure. This is published in Crump et al (PLoS Comp Biol, 2021) and is summarised in the Methods of this paper. We have made many citations to this original fitting paper throughout the main text (e.g. lines 40, 51, 62, 107, 165, 291, 301 and 312). The focus of this paper was on the future strategies and corresponding predictions for transmission and case reporting under them rather than the statistical fitting methodology presented previously. We believe that this paper is far more policy-facing compared to the previous article as the results can provide strong guidance for decision making.
3) There is no model comparison. It's unclear whether simpler models can work better. For example, complex models may lead to overfitting. So it's unclear whether the projection is robust or not.	We agree that model comparison can be a valuable exercise to protect against overfitting. Firstly, we wanted to use a mechanistic model as this enables us to make predictions even under new interventions which haven't occurred before; this isn't possible with a purely statistical model. Furthermore, in this article we utilised a previously developed model (first published in

2015) and variants of this have been carefully adapted based on case data to explore different model structures for DRC and other gHAT-endemic regions previously. Of particular note, the original model paper (Rock 2015 P&V (M1-M7 in Yasa Bonga and Mosango)) looked at 7 different model variants, ranging from a basic Ross-Macdonald type model (M1) to a risk-structured model with animal transmission to assess which had best statistical support given the longitudinal case data.

Subsequently, further analysis was conducted in the same vein for a slightly updated model (Mahamat 2017 PLoS NTD (M1-M8 in Mandoul, Chad)) which also included validation steps to remove data and compare predictions. In these analyses we consistently found strong support for Model 4 (high- and low-risk structure with non-participation in active screening from high-risk individuals).

Finally, Castaño et al (PloS NTD, 2020) examines whether or not there are improvements to passive screening over time by comparing fits to staged case data and finds that the trends in parts of DRC are not possible to replicate without improvements in passive screening. All of these details are given in full in the original fitting paper and provide strong justification for why simpler models cannot provide such well-matched outcomes when compared to data from various geographies.

We agree that other readers may have the same question and so we added the following in the discussion section (line 165 & 169): “Using a mechanistic SEIR-type model not only provides parameter estimates that describe the dynamics of infection but also allows us to simulate future dynamics under various strategies, including those not conducted in the health zone previously. Other simpler model variants published elsewhere are not considered in this study as none of them can achieve good health zone-level fits to the WHO HAT Atlas data aggregated by health zone and year [22, 25, 29].”

	We also added the following in the methods section (line 272): “Previous modelling studies explored several model variants for some health zones within the former Bandundu province in DRC and a focus in Chad. They showed that a simpler model variant (“Model 1”) which is a Ross-Macdonald style model was unable to fit the case data well in DRC or Chad and therefore concluded that heterogeneous risk of human infections, participation in AS structures and improvement in passive screening are essential to perform a good model fit to observed longitudinal human case data [22, 25, 29].”
(4) There is no discussion about the identifiability of model parameters. For example, the authors may need to generate some simulation data, and see if the inference method can work well to identify the true parameters. Otherwise, it’s hard to see whether the inference and projection were done correctly.	Fitting was done in a previously published paper (Crump et al, PLoS Comp Biol, 2021) and therefore the results of this are reported within that manuscript and SI. Details of posteriors for all health zones can be found in the linked graphical user interface (GUI) https://hatmepp.warwick.ac.uk/fitting/v2/ which is referenced in the original paper. This also gives the reader a sense of how the model is able to fit the longitudinal data in each of the different health zones, split by active and passive detections. Furthermore visualisations of the fitting for years 2000-2016 can be viewed in the projections GUI (https://hatmepp.warwick.ac.uk/projections/v2/) by adjusting the date range in the “active detections” and “passive detections” tabs which gives a sense of how well the model fits in each health zone. We added the following in the results section (line 50): “Our gHAT model, a deterministic mechanistic SEIRS-type model, was independently fitted to longitudinal human case data from 2000-2016 in 168 health zones in DRC by MCMC methods [30]. We used parameter estimates from our previously fitted gHAT model (posteriors are available at https://hatmepp.warwick.ac.uk/fitting/v2/ and simulated active and passive cases in 2000--2016 can be also viewed at https://hatmepp.warwick.ac.uk/projections/v2/)

	to simulate forward projections in 168 health zones under four main strategies: two medical-only strategies which comprise of active and passive screening (MeanAS and 40%AS), and two medical strategies with supplemental vector control from 2020 (MeanAS+VC and 40%AS+VC). In the GUI we include 12 additional projections for each health zone for alternative levels of vector control (60% and 90% tsetse reduction, as well as our default of 80%) and active screening at the historical maximum observed for each health zone (HistMaxAS).”
(5) It’s unclear whether the authors considered the reporting bias to detect cases. For example, different locations in DRC may have different reporting rates. So that some parameters related to the reporting system are needed.	This was included in the original model fit and a description is given in the original paper. We account for both variable annual active screening coverage within each health zone and also geographical differences in time to detection through passive screening through our fitting procedure. We have adjusted the methods section in our text (line 306) to make this as clear as possible: “For each health zone we fitted, we estimated model parameters that are likely to geographically vary across regions including: basic reproduction number, proportion of low-risk humans, relative bites taken on high-risk humans, and reporting bias such as passive treatment rates for both stages, passive reporting proportion and active screening diagnostic specificity (Supplementary Table 2).” We also note that we account for health zone-specific screening trends based on historical data (line 300): “In a previous publication we used the gHAT case data and annual screening numbers in 168 health zones from the WHO HAT Atlas to individually fit the gHAT model to health zone specific trends from 2000–2016 [30]. The data are aggregated by location, year, surveillance type and diagnosed disease stages (stage 1, stage 2 and stage unknown). Location was defined by the available geolocation and geographical identifier information, while surveillance type

	was either active or passive screening [12, 45, 46].“
(6) The model structure is simply considered as a single population for each health zone in DRC. It’s not clear whether human mobility and the movement of the vector will affect the distribution of cases.	It is correct that the model does not account for inter-health zone movement of people or tsetse, but the high-/low-risk structure does incorporate some heterogeneity of risk within health zones. Our model fits match the data well, although we take this point and have added an additional section in the discussion about risks to places that achieve EOT earlier but are neighbouring higher prevalence health zones. We have slightly rearranged our discussion to emphasise how we would expect movement to alter our results (see paragraph starting on line 224): “Host or vector movement may play some role in spreading infection. Health zones that achieve EOT earlier but are neighbouring higher prevalence locations have a higher chance of resurgence due to human mobility. The typical size of health zones in DRC is much bigger than the spatial scale of tsetse movement, although epidemiological foci may overlap health zone boundaries and could impact predictions. A stochastic model applied at a smaller spatial scale is more suitable for addressing issues related to chance events and their impact on EOT [28].”
(7) Are you fitting each location separately or fit all locations together using a single likelihood?	We fit to each health zone separately, however there are some former province-specific priors for passive detection improvements over time based on provincially aggregated data that were available. This is noted in the methods section (line 301) “individually fit the gHAT model to health zone specific trends from 2000–2016 [30]” and more thorough details are given in the original manuscript (Crump et al, 2021).

Reviewer 2

“This is an interesting and useful paper that uses an existing and well published model framework at a national scale to try to give information about how to prioritise HAT control resources.

For context I have been conducting the review of this paper whilst considering possible areas outside former Bandundu that could benefit from vector control, so being able to review this paper has been very useful, but has also raised a lot of questions about the model. At a high level I have misgivings about presenting the results purely in terms of time to elimination of transmission. I think that a broad piece such as this going into a high impact journal the results should be primarily presented in terms of public health impact with elimination of transmission as a secondary output. Elimination of transmission is a target set out by funders and stakeholders, but is merely a concept to the teams implementing controls on the ground who measure impact in terms of public health impact. “Why are you proposing we work in this health zone?” ANS “Because the model says that it will not achieve elimination of transmission before 2040” is quite a weak response. “Why should we work in this health zone” ANS “Because it will prevent XXX deaths” carries a bit more punch. Elimination of transmission is certainly important but must be considered alongside the public health impact.

Also at a high level. You don’t make much of it, but you results give some strong indications about the efficacy of active screening to drive elimination. I think that people will be surprised by how poor are the positive predictive values of active screening detections and these are only going to get worse. This poor PPV is part of the reason why active screening is predicted to play relatively little part in the targets. It is a surprising result and as a consequence needs more attention – more below.

In terms of deciding where it may be possible to site vector control activities one’s eye is drawn on your maps to the massive clumps of dark red in Maniema and Katanga. One’s eye is drawn here because they are not areas that have a high HAT burden and so were not on the radar for many. Yet they are predicted to fail to make the elimination targets which some higher incidence ZS are predicted to hit. Because it is surprising and unexpected does not make it wrong but does require additional scrutiny – one needs to be very sure before directing resources to low-burden areas.”

Author response:

We thank the reviewer for the constructive criticism of our submitted manuscript and have taken these suggestions on board. In particular we have addressed the disease burden question by now making more of the model outputs corresponding to deaths from gHAT. We also modified our four main strategies to make a fairer comparison across health zones with high coverage active screening (40% coverage). We have addressed specific changes to the manuscript to address these and other remarks below.

Major corrections	Author response
1. Please make the results clearly available in terms of public health impact (this could be as DALYs, mortalities averted, infections	In the companion paper (also under review for Nature Comms and preprinted on MedRxiv https://www.medrxiv.org/content/10.1101/202

averted or YLLs). I think that this is useful for people on the ground and those planning activities, particularly as from your model disease burden and time to elimination are not well correlated and we don't want people to conflate them

0.08.25.20181982v2) we focus much more on DALYs averted as part of our health economic analysis. However, we agree that more could be made of the disease burden outputs we generate alongside these projections - specifically we now present our estimates for deaths outside of health care which make up most of the DALY burden estimates for gHAT.

In particular for Figure 5 we changed the y-axis from PEOT (probability of elimination of transmission) to total deaths to select the priority list using our model. We did actually find some negative correlation between estimated burden and PEOT by 2030 and the top 10 health zones in terms of burden all had <40% probability of reaching PEOT by 2030. We revised the text (line 143):

“The symptoms of gHAT are generally mild and non-specific before trypanosomes cross the blood-brain barrier, however progression to more severe symptoms and then death is the outcome when infected humans are missed by AS and never identified in PS. Thus, the disability-adjusted life years (DALYs) of gHAT, a broad measure of overall disease burden, are mainly comprised of deaths outside the health care system. Health zone maps of total deaths under MeanAS strategy (Figure S3) and deaths averted under intensified interventions (Figure S4) are available in the Supplementary Results section. By identifying health zones which have greater than 50 total deaths predicted in 2017-2030 under the 40%AS strategy we compiled a priority shortlist of health zones in the former Bandundu province where VC implementation is practically feasible and highly recommended by mathematical modelling: Kwamouth, Masi Manimba, Bokoro, Bagata, Mushie, Kimputu, Mokala, Bulungu, Nioki, and Kenge (as shown in Figure 5).”

We added in the death outcomes (estimated deaths outside healthcare) into the GUI so we can see the impact of strategies on disease burden as well as progress towards the EOT target. We have also added two figures --

	total deaths under MeanAS strategy (Figure S3) and deaths averted with intensified strategies (Figure S4) in the SI to highlight this public health impact.
2. meanAS and maxAS. These are based on historical precedent, but can often vary 10x between health zones. meanAS I don't have a big concern with because it is showing the business as usual scenario. maxAS is a problem. Unless the reader really digs they could be forgiven for assuming that maxAS represents the same screening effort across all health zones, not potentially a 10x difference. Furthermore, it is really difficult to find out what maxAS represents in terms of screening effort. Why when VC is based on a theoretical target can we not have a theoretical target for AS to compare between health zones. Why would it be so difficult for a ZS to raise its screening from a previous peak of 5% to 10%, particularly when one considers that active screening can be scaled up or redeployed much faster than vector control resources. Hence, when I look at these ZS in Maniema and Katanga I wonder whether they are a particular problem area for HAT that really requires vector control or whether they are just areas that have never received much active screening	We added strategies with 40% AS to our simulations and now present them (40%AS and 40%AS+VC) in the main text. We changed the name of MaxAS to HistMaxAS to highlight that the maximum values are determined by the historical data. We agree that HistMaxAS is harder to represent as it's not a real limit in terms of screening effort and moved results from HistMaxAS and HistMaxAS+VC to the SI. We added maps of deaths averted under intensified strategies in the SI (Figure S4) showing that health zones in Kasai Occidental, Kasai Oriental, Katanga and Maniema benefit more from increasing AS (see line 110 in the SI): "Only a few health zones have high deaths averted under HistMaxAS because (1) the historical maximum AS coverage is too low to pick up many infected individuals in some health zones, particularly in the east of the country; (2) in health zones where the mean AS coverage is already good (e.g. in former Bandundu province), there can be marginal benefit from increasing coverage as our simulated strategy never reaches the high-risk individuals through AS. By comparing deaths averted under the 40%AS strategy to that under the HistMaxAS strategy, large numbers of deaths are predicted to be averted in Kasai Occidental, Kasai Oriental, Katanga and Maniema provinces."
3. Homogenous risk. Your model assumes that the risk and incidence is spatially homogenous within the ZS, albeit with high risk individuals that are not actively screened. This seems fine in the higher incidence health zones in which generally the majority of the population is at risk (albeit likely with local variations in the magnitude of the risk). My concern is with the lower incidence health zones where	We are aware of the limitation of assuming homogeneous case distribution and intervention efficacies in our work, nevertheless we believe that this model operating at a health zone level captures the historical infection dynamics well and our calibration of the proportion of the total population at high-risk and the relative risk of the high-risk group allows us to mimic geographical differences between health zones of different prevalences.

you have had to assume that the low incidence is homogenously smeared across the ZS and so the whole population is at low or very low risk. In some ZS this may be the case, but in reality it is more likely to be highly focalised with areas of high risk populations and larger portions of the population at very low or negligible risk. Where I think that this matters is that it contributes to the low efficacy of AS. If maxAS is 5% in one low incidence ZS and 50% in a higher incidence ZS then you are modelling that translates to 5% and 50% of the populations at risk getting screened. However, active screening is targeted to those villages that are perceived to be at risk (or at least have a recent history of HAT) – your co-authors from PNLTHA could expand on this. So, considering this, if in our low incidence ZS only 10% of the population is at risk then the screening will be targeted at this population, so the effective screening rate would be 50%.	We agree that the same screening coverage of X% in two health zones would not translate to the same effective coverage, although it would represent similar levels of resource use (i.e. similar number of people screened). A health area or village scale model could account for within health zone heterogeneity in risk in a more robust manner, however further work is needed to assess in which regions there would be sufficient reliable data to calibrate models to these smaller spatial scales and overcome other computational challenges associated with fitting a model to around 10 times as many locations (for health areas) and greater levels of stochastic noise. Despite its complexity, smaller scale modelling is clearly a priority for the future but it is beyond the scope of this present paper. We added the following sentences in the discussion section to emphasise how we would expect case clustering and low incidence to impact our model (line 247): “We did not consider the geographical distributions of reported cases within health zones in our model as our model already produced fits well-matched to data. As a result, it is plausible that the relative risk of high-risk humans (r) is underestimated and the proportion of low-risk humans (k_1) is overestimated. Model fitting and projections at smaller scales would provide more realistic impacts on the efficacy of AS and VC when infections are highly clustered in smaller areas or when incidence becomes very low and trigger the cessation of AS. To achieve smaller scale analyses in the future there are a range of computational challenges that would need to be overcome, such as an order of magnitude increase in the number of locations to be fitted to, and greater levels of stochastic noise in the data. In some smaller scale regions there may simply not be sufficient historical data points to allow for robust fitting of the model.”
4. (I think) Your estimate of specificity is based on an estimate of the proportion of all	We realise from this comment that we weren't clear enough about how we dealt with false positives in the model. We assume that all

cases that are false positives. This may not impact the results or the estimates of specificity, but to do this correctly I think that you need to separate HAT cases by stage. HAT cases in stage 2 are much less likely than cases in stage 1 to be false positives. This is because to be a stage 2 HAT cases there must be at least 2 of 4 or 5 positive indicators. To be stage 2 a case must have ((positive parasitology and / or strong clinical signs and / or CATT dilutions) AND (trypanosomes in CSF and / OR elevated WBC count)). It is not without error, but much less likely to produce false positives than stage 1 which requires just one indicator. In my view specificity would be better fitted using just stage 1 cases and assuming that all stage 2 cases all true positives.	false positives would be assigned to be stage 1, this has also been used across all the other Warwick gHAT model DRC publications. We added this information to the following sentence (line 309): “The fitted model also takes into account previous advances in medical, diagnostic, and control systems and assumes all false positives found in active screening are assigned to stage 1.”
5. You need to present the fitted parameters for all ZS in supplementary information or through the GUI. If we are thinking about these timelines to elimination as a linear equation then elimination year (y) is $y = a + bx$ where a is the number of HAT cases at the start of predictions x is time and b is a function of HAT control and the fitted epidemiological parameters. So, to use an example we can compare Samba ZS in Maniema to Bagata ZS in Bandundu. Our value for a in Samba is 27 but it is 47 in Bagata. However, under the basic scenario y is 2039 for Bagata but > 2040 for Samba. This must mean that the differences lie within b. Now it could be down to the meanAS parameter, or it could also be within the fitted parameters. Hence the reader needs to know where the cause lies.	We added the following sentences in the results section to clarify that the elimination years are highly dependent on the fitted epidemiological parameters. A GUI link is now also provided for readers to easily access the posteriors of fitted parameters. See paragraph starting on line 101: “Case reporting has been the primary but indirect measure for underlying transmission. So one may expect different health zones with the same number of reported cases very likely to have different predicted years of EOT and certainty of EOT by 2030. Lusanga and Mosango health zones are geographically connected and both had 13 total reported cases in 2016. Our model predicts EOT to happen in 2029 in Lusanga and 2027 in Mosango under 40%AS strategy and the probability of EOT by 2030 are 60% and 83% respectively. These differences come from underlying epidemiological variation such as relative risk of high-risk people, tsetse density or time to detection through passive screening (linked to health facility coverage and attendance). The explanations for some

	of these differences are explored in our fitting paper [30] and the posterior distributions of the parameters can be found through the accompanying GUI, https://hatmepp.warwick.ac.uk/fitting/v2/ .”
--	---

Minor comments	
Line 3. The word "yet" is erroneous	Deleted.
Line 4. I would remove the parenthesis with (potentially illusive), either way, I think you mean elusive	Done.
Lines 2-7 (particularly 5-7 really require some evidence to support them	We have inserted additional references.
Lines 13-16. A long and clunky sentence. Give the reader a hand here	Amended.
Line 21. English language pedantry. Replace "most" with "greatest number of..."	Done.
Line 24. Is it? Where it will end up depends upon the direction of travel as the starting point, so without a reference this is a weak statement , particularly in the context of a disease which has little control in some countries.	We replaced “the most critical” with “a critical”.
Line 26. "has" not "have" and can you give us more information here on the distribution of populations of HZs. As worded, this suggests that they are quite uniform in size which I didn't understand to be the case, but may bely my ignorance. You should also add some information for the reader that public health care is manages at the HZ level	Corrected. We revised the sentences to provide the reader with more information: “In order to assess EOT feasibility, this study focuses on quantitative forecasting of gHAT across the endemic health zones in DRC to examine if, how, and when EOT could be expected under strategies based on currently available tools. Health zones are the administrative units at which public health care is managed and each has a population of between 29,010 and 613,072 people (the median size is 157,338).”
Line 32. How are the 168 HZs defined as endemic?	We added: “These [168] health zones are ones considered to be endemic during the

	2000--2016 period, having reported cases in a minimum of 5 years.”
Introduction generally. Is very light indeed on references 13 in total of which 7 are references to your own group's modelling work. There are a lot of big statements without any support and a lot of HAT-specific terms and concepts which the wider reader may not be familiar with (such as VC and AS) which the reader may not be familiar with in the HAT context and gets no referential support.	Thank you for pointing this out. We have now backed up such statements with appropriate references. We have also added the following sentences to give the reader more upfront context about gHAT interventions: “Advances in treatment have transformed the once toxic intravenous treatment regime into an oral cure for most patients, however confirmation of the parasite is currently still a requirement before the drugs can be administered [18,19]. This means that infected individuals must be identified either by self-presenting at facilities with gHAT diagnostics due to symptoms or by specially-trained mobile screening teams targeting villages with recent case reporting [16]. A complementary intervention of tsetse control has not been used widely to date, but has been shown to be effective at rapidly reducing vector populations in some gHAT-endemic regions where it has been implemented [20-22].”
Lines 226-231. See my general comment on heterogeneity	Please see our response above.
Line 255. Add a space after future	Added.
Table S2. Are the parameters below dchange supposed to be there?	The aggregated province-level data shows strong evidence of improvements to the passive surveillance system (the changes to the proportion of passive cases identified in stage 1 compared to stage 2) during 2000--2012 in the former Bandundu and Bas Congo provinces. We used logistic functions with three parameters (the relative amplitude (η_{Hamp} and γ_{Hamp} for stage 1 and 2 respectively), the speed (d_{steep}), and the time point of improvement (d_{change})) to describe the change of passive detection in time. These parameters are considered as fitted parameters because we believe they vary from health zone to health zone. We don't include these parameters in the fitting for other former provinces as there is no similar evidence that such improvements occurred based on historical data and this is

	why we don't provide estimates for these parameters in Tandala (in former Equateur province).
Figure 1 (& GUI). Please mark the point at which AS specificity goes up to 100%	We added a colour background in figure 1, and figures S1-S2 to indicate the point at which AS specificity goes up to 100%. For GUI, we added a new table to provide the values of the median and the 95% CI of our predictions on the year that AS specificity becomes 100% under different strategies.

REVIEWERS' COMMENTS

Reviewer #1 (Remarks to the Author):

The authors improved the manuscript, although there seems no track change that could help understand their detailed revisions. After reading this revised version, I have several comments as follows:

1. The data used to fit the model is from 2000 to 2016, which is before the COVID-19 pandemic. However, the forecast is for years up to 2050. It is unknown whether the transmission dynamics of gHAT have a substantial change during and after the COVID-19 pandemic. Readers will ask whether it is suitable to train the long-term forecast model using old data collected before the COVID-19 pandemic.

2. The number of health zones increased from 57 health zones in Rock et al (Epidemics, 2017) to 168 health zones in this manuscript could NOT be regarded as a major improvement in the methodology of mathematical modeling and epidemiology. The revised manuscript mainly performs a mathematical modeling exercise by applying their existing published model and parameters to a larger dataset. For high-impact journals such as Nature Communications, readers may wish to learn more than a mathematical modeling exercise.

For example, the 168 health zones considered in this study only account for around 1/3 areas of the whole DRC (Figures 3 and 4). When comparing the new results from this study with those published previously, could you identify some underlying mechanism or factors driving the patterns in the observed data? Could you provide some predictive model to extrapolate the results of 168 health zones to the whole DRC?

3. This study is trying to predict the long-term disease outcome of gHAT (up to 2050). However, there will be many unknown changes in the next 30 years, such as the demographic change affecting the risk of exposure for humans and the environmental changes affecting the distribution of vectors. Such large uncertainties will make long-term forecasts meaningless.

Reviewer #2 (Remarks to the Author):

The authors have done considerable work and much improved the manuscript. well done

EOT versus public health measures. Following the first review, the authors have added more on public health measures, but I still feel that the headline should be the public health impact rather than the rather arbitrary elimination of transmission goals. This is particularly the case as the authors clearly demonstrate that the elimination of transmission targets are unreachable, which does leave something of a quandry of "well how do we prioritise resources". However, this is not a critical point to address.

As a user of the outputs I would still treat them with great caution due to the discussed assumption of homogenous risk when the focus may cover a small part of the health zone. But the outputs are useful guidance

Very minor:

Lines 130-142. The grammar in this paragraph goes a little off the rails

REVIEWER COMMENTS

Reviewer #1 (Remarks to the Author):

The authors improved the manuscript, although there seems no track change that could help understand their detailed revisions. After reading this revised version, I have several comments as follows:

Author response:

Comment	Author response
No track change file	We apologise a track changes file was not uploaded last time and include a file here which shows the current file compared to the original submission.
1. The data used to fit the model is from 2000 to 2016, which is before the COVID-19 pandemic. However, the forecast is for years up to 2050. It is unknown whether the transmission dynamics of gHAT have a substantial change during and after the COVID-19 pandemic. Readers will ask whether it is suitable to train the long-term forecast model using old data collected before the COVID-19 pandemic.	Thank you for pointing this out. We agree that readers may have this question and indeed, for many disease control programmes, the pandemic has substantially impacted them. In the case of gHAT control in DRC we know that most of the programme was able to continue functioning as previously, however screening coverage in 2020 was lower than originally planned. Our model already explores the impact of different levels of active screening coverage, and whilst the exact coverage for all years is unlikely to be represented exactly, our projections are designed to give an overall qualitative picture of how coverage will impact transmission and reporting as well as specific quantitative outputs for the selected strategies. Our group explored the expected impacts of interruptions to different parts of the gHAT programme in DRC in different prevalence settings and concluded that if only active screening was impacted for a short duration the main impact would be on case reporting during this interruption period, but there would be limited effects on disease burden or transmission due to the slow progressing nature of gHAT disease (Aliee et al., Trans of the Royal Soc Trop Med & Hyg, 2021). We found that if both active and passive screening was completely interrupted for several years then we may expect delays to EOT but we now know that this worst-case scenario did not occur. Currently the detailed data we would need to

include in DRC-wide updated fitting and projections for the health zone level are not available for 2020 onwards. Due to the interruptions to active screening only, we believe that our projections remain reasonable, despite the COVID-19 pandemic.

We previously alluded to this in our discussion section (line 221), but have now expanded this by adding:

“Our model framework is flexible and could be used to predict the impact of unexpected future changes by estimating how they could alter observable (i.e. reported cases and deaths) and unobservable variables (i.e. new infections); in the present climate of the COVID-19 pandemic and recent Ebola outbreaks in gHAT-endemic parts of DRC, this is particularly relevant and could provide support in planning whether subsequent gHAT interventions should be altered due to unforeseen interruptions. Other work by this modelling group specifically analysed the potential impact of different COVID-19 interruptions to the gHAT programme in DRC and concluded that short interruptions to active screening coverage only were unlikely to result in delays to EOT due to the long timescale of disease progression in humans [33]. If longer interruptions occurred and impacted both active and passive screening then we would expect delays of similar duration to the interruption itself, however it is now known that this worst-case scenario did not take place and the main impact on the DRC programme was lower coverage of active screening during 2020.”

In addition we describe in the discussion how future work could update our model predictions as more data become available which might include during the COVID-19 period (line 218):

“When the new data from 2017 onward becomes available, we will be able to use it to validate our model by changing assumed AS coverage to actual numbers of people screened and then comparing the predicted active and passive cases to reported cases. Subsequent re-fitting to the recent case data would further refine model predictions presented here and is an important step in the continuous process of modelling to support policy under NTD-PRIME principles [35] (Supplementary Table 3).”

2. The number of health zones increased from 57 health zones in Rock et al (Epidemics, 2017) to 168 health zones in this manuscript could NOT be regarded as a major improvement in the methodology of mathematical modeling and epidemiology. The revised manuscript mainly performs a mathematical modeling exercise by applying their existing published model and parameters to a larger dataset. For high-impact journals such as Nature Communications, readers may wish to learn more than a mathematical modeling exercise.

For example, the 168 health zones considered in this study only account for around 1/3 areas of the whole DRC (Figures 3 and 4). When comparing the new results from this study with those published previously, could you identify some underlying mechanism or factors driving the patterns in the observed data? Could you provide some predictive model to extrapolate the results of 168 health zones to the whole DRC?

This paper does not simply represent an increase in the number of health zones considered, but also represents a considerable increase in model complexity (matching more of the underlying biology) and greater robustness in terms of statistical methodology. In our description of the paper we wished to highlight that we were not simply presenting a small technical improvement but were extending the scale of our analysis in order to address a range of issues that are important to national policy-makers and donors. Many of the technical advancements are outlined in Crump et al., PLoS Comp Biol 2021), but here we focus on the potential gap between the current strategy and the WHO goal of EOT to support PNLTHA-DRC, their donors and partners in decision making towards lower disease burden and transmission.

This is the first time projections have been available for all health zones of the DRC reporting more than 10 annual recordings (active screening or passive detections) during 2000-2016. The results we present here therefore cover all areas under control by PNLTHA-DRC. The 168 analysed health zones account for 98% of reported gHAT cases from DRC in the past 17 years.

In the original data, out of the 516 health zones of DRC only 290 outside of urban Kinshasa report any data (either an active screen or any cases). We did not fit the model in places where we could not robustly estimate model parameters from the available data (i.e. < 10 data points). In Kinshasa we also excluded the urban health zones (30) with case reporting as these are not regions with tsetse habitat and therefore transmission does not occur in these regions, rather these cases will be from people diagnosed in the region but infected elsewhere.

Most health zones in DRC (210) reported no HAT cases during the data period and are believed to not be regions with transmission. HAT activities such as active screening have been conducted in health zones which have had historical cases. Whilst we cannot say with complete certainty that these 210 health zones have no transmission, we believe that it remains unlikely that persistent transmission has occurred in these locations

	without being reported. Therefore whilst we are not willing to make quantitative predictions for the health zones coloured grey on our maps using our model, we believe that these largely represent regions without transmission. If one wanted to assess the transmission potential across different geographies with no case reporting, the only option would be to assess geographical indicators correlated with transmission potential such as suitable tsetse habitat (rivers and vegetation), etc. Similar work has been done elsewhere (Tirados et al., PLoS NTD, 2020) but uses a very different modelling approach and data. It is also worth emphasising that geographical/habitat suitability for transmission does not mean that transmission currently occurs in those regions. We have elaborated on this further in the methods section to make this clear to readers who are not familiar with the distribution of gHAT in DRC (line 328): “Cases of gHAT are not reported across all of the DRC (only 290 out of 516 health zones had any data for the 2000-2016 period), and some health zones have very little case reporting or screening activities (122 health zones of the 290 reporting any data). We do not include these locations in either fitting to the historical data or future predictions and they are coloured grey on corresponding maps. Whilst we are not willing to make quantitative predictions for the health zones coloured grey on our maps using our model, we believe that these largely represent regions without transmission.”
3. This study is trying to predict the long-term disease outcome of gHAT (up to 2050). However, there will be many unknown changes in the next 30 years, such as the demographic change affecting the risk of exposure for humans and the environmental changes affecting the distribution of vectors. Such large uncertainties will make long-term forecasts meaningless.	We agree that a challenge with any epidemiological modelling is making long-term predictions in the face of uncertainty in a range of factors. Our policy recommendations in this article are based on the expected infection, disease burden and reporting dynamics in the next decade, for which we have much higher confidence. However we present simulations up to 2050 to be able to provide illustrative outcomes beyond 2030, in particular whether we expect the EOT goal to be missed by a few or by many years. Whether or not elimination is expected to occur before 2030, slightly beyond 2030 or after 2040 is an important

metric of value which our projections can be used to assess. We believe that whilst it would be unwise to expect our model to be accurate at 2050, the projections still demonstrate the trends we might expect given the specific strategies and current situation.

In short, such projections are used to consider if present-day activities are on the right track for the stated goals of the stakeholder, even without abrupt changes in other factors. If the answer is no, then that is enough to warrant policy changes. Whether we are right to within 1% of cases by 2050 is not necessary for modelling to prove useful.

We have added the following to our discussion section on limitations of model projections for long time horizons (line 209):

“Whilst our policy recommendations are based on the expected infection, disease burden and reporting dynamics in the next decade, we also report long-term projections up to 2050. Making long-term predictions is always a challenge for epidemiological modelling in terms of uncertainty in a range of factors (e.g. demographic and environmental changes). However, presenting simulations up to 2050 provides illustrative outcomes beyond 2030, such as whether we expect the EOT goal to be missed by a few years or if it might be totally infeasible. Whether elimination is expected to occur before 2030, slightly beyond 2030, or after 2040 is a metric of value which our projections can be used to assess. Our projections are used to consider whether current strategy is on the right track to meet policy objectives or whether a change in strategy is warranted; high levels of accuracy in model projection at 2050 are not necessary for modelling to prove useful in these assessments.”

Reviewer 2

The authors have done considerable work and much improved the manuscript. well done

Author response:

Comment	Author response
EOT versus public health measures. Following the first review, the authors have added more on public health measures, but I still feel that the headline should be the public health impact rather than the rather arbitrary elimination of transmission goals. This is particularly the case as the authors clearly demonstrate that the elimination of transmission targets are unreachable, which does leave something of a quandry of "well how do we prioritise resources". However, this is not a critical point to address.	We agree that focusing on disease burden metrics in addition to EOT goals is rational given the article's conclusions. We retain the metric of EoT since this is a target agreed upon by a large number of stakeholders (including WHO), however for the purpose of prioritisation we have made further efforts to boost disease burden discussion in our text. Our new maps and outputs produced in response to this reviewer's suggestion in the previous round of revisions already show these outcomes, and in this second revision we have paid more careful attention to details in the text. We added more mention of disease burden, cases, and deaths throughout the main text (e.g. lines 25, 31,45, 103, 104, 127, 178, 180, 233, 241, 250)
As a user of the outputs I would still treat them with great caution due to the discussed assumption of homogenous risk when the focus may cover a small part of the health zone. But the outputs are useful guidance	We have elaborated on the limitation of homogeneity assumption in the discussion section to make this clear to readers (line 269): "Disease foci (transmission pockets) and intervention areas may only cover small parts of the health zone. We did not consider the geographical distributions of reported cases within health zones in our model. Although our model already produced fits well-matched to data, we expect that the relative risk of high-risk humans (r) is underestimated and the proportion of low-risk humans (k_1) is overestimated due to the assumption of homogeneity within health zones."
Lines 130-142. The grammar in this paragraph goes a little off the rails	Amended.